# Cognitive Processing Therapy or Relapse Prevention for comorbid Posttraumatic Stress Disorder and Alcohol Use Disorder: A randomized clinical trial

Tracy L. Simpson[1,2‡]*, Debra L. Kaysen[2,3‡], Charles B. Fleming[4], Isaac C. Rhew[4], Anna E. Jaffe[5], Sruti Desai[4], Denise A. Hien[6], Lucy Berliner[7], Dennis Donovan[8], Patricia A. Resick[9]

1 Center of Excellence in Substance Addiction Treatment and Education, VA Puget Sound Health Care, Seattle, WA, United States of America, 2 Department of Psychiatry and Behavioral Sciences, School of Medicine, University of Washington, Seattle, WA, United States of America, 3 Department of Psychiatry and Behavioral Sciences, Stanford University School of Medicine, Stanford, CA, United States of America, 4 Center for the Study of Health and Risk Behaviors, Department of Psychiatry and Behavioral Sciences, School of Medicine, University of Washington, Seattle, WA, United States of America, 5 Department of Psychology, University of Nebraska-Lincoln, Lincoln, NE, United States of America, 6 Department of Clinical Psychology, Graduate School of Applied and Professional Psychology, Rutgers University, Center of Alcohol & Substance Use Studies, Piscataway, NJ, United States of America, 7 Harborview Center for Sexual Assault and Traumatic Stress, University of Washington, Seattle, WA, United States of America, 8 Alcohol and Drug Abuse Institute, Department of Psychiatry and Behavioral Sciences, School of Medicine, University of Washington, Seattle, WA, United States of America, 9 Department of Psychiatry and Behavioral Sciences, Duke Health, Durham, NC, United States of America

‡ TLS and DLK are Joint First authors to this work.
* Tracy.Simpson@va.gov

**Data Availability Statement:** Data are available on openICPSR (DOI: 10.3886/E182022V1).

## Abstract

### Objective

To compare a Posttraumatic Stress Disorder (PTSD) treatment (Cognitive Processing Therapy; CPT), an Alcohol Use Disorder (AUD) treatment (Relapse Prevention; RP), and assessment-only (AO) for those meeting diagnostic criteria for both PTSD and AUD.

### Method

Participants with current PTSD/AUD (N = 101; mean age = 42.10; 56% female) were initially randomized to CPT, RP, or AO and assessed post-treatment or 6-weeks post-randomization (AO). AO participants were then re-randomized to CPT or RP. Follow-ups were at immediate post-treatment, 3-, and 12-months. Mixed effects intent-to-treat models compared conditions on changes in PTSD symptom severity, drinking days, and heavy drinking days.

### Results

At post-treatment, participants assigned to CPT showed significantly greater improvement than those in AO on PTSD symptom severity (b = -9.72, 95% CI [-16.20, -3.23], d = 1.22);

**Funding:** • Specific grant numbers: R01AA020252-01 • Initials of authors who received each award: DK & TS • Full names of commercial companies that funded the study or authors: No commercial companies; study was funded by the National Institute on Alcohol Abuse and Alcoholism • Initials of authors who received salary or other funding from commercial companies: None • URLs to sponsors' websites: https://www.niaaa.nih.gov/ The funders had no role in study design, data collection and analysis, decision to publish, or preparation of the manuscript.

**Competing interests:** Dr. Kaysen is a co-author of a book on Cognitive Processing Therapy published by Elsevier for which she receives royalties. In addition she has conducted clinical workshops on Cognitive Processing Therapy for which she has received speakers fees, which could constitute a conflict of interest. Dr. Resick is a co-author on the Cognitive Processing Therapy treatment manual for which she receives royalties and she conducts clinical workshops on Cognitive Processing Therapy for which she receives speakers' fees, which could constitute a conflict of interest. The other co-authors have no conflicts of interest to declare pertinent to this submission. This does not alter our adherence to PLOS ONE policies on sharing data and materials.

the RP and AO groups did not differ significantly on PTSD. Both active treatment conditions significantly decreased heavy drinking days relative to AO (CPT vs. AO: *Count Ratio [CR]* = 0.51, *95% CI* [0.30, 0.88]; RP vs. AO: *CR* = 0.34, *95% CI* [0.19, 0.59]). After re-randomization both treatment conditions showed substantial improvements in PTSD symptoms and drinking between pre-treatment and post-treatment over the 12-month follow-up period, with RP showing an advantage on heavy drinking days.

## Conclusion

Treatments targeting one *or* the other aspects of the PTSD/AUD comorbidity may have salutary effects on both PTSD and drinking outcomes. These preliminary results suggest that people with this comorbidity may have viable treatment options whether they present for mental health or addiction care.

## Trial registration

The trial is registered at clinicaltrials.gov (NCT01663337).

## Introduction

Over half of community-dwelling individuals with lifetime Posttraumatic Stress Disorder (PTSD) also have lifetime Alcohol Use Disorder (AUD), and approximately 12% of individuals with lifetime AUD have lifetime PTSD [1]. Among those seeking substance use disorder (SUD) treatment, approximately 60% have comorbid PTSD [2, 3]. Current conventional wisdom is that combined or integrated treatments addressing both aspects of the comorbidity are preferable for those with PTSD/SUD [4–8]. However, there may be practical reasons for utilizing single-focus treatments. A recent review describes numerous barriers to receiving integrated care for dual diagnoses [9] and consistent with the wider comorbidity literature [10, 11], recent epidemiologic research indicates that most individuals with PTSD/AUD report receiving only mental health-oriented care (i.e., not care for their AUD; 58%) with less than a fifth reporting receiving care for both issues [1]. From a public health standpoint, should randomized clinical trials reveal that single-focus treatment options successfully address both aspects of PTSD/AUD, some lower resource mental health and substance abuse clinics may wish to improve the fidelity of their single-focus offerings rather than invest in training staff on new treatments.

Although typically a control condition in PTSD/SUD treatment studies, cognitive-behavioral treatments (CBT) for SUD, such as Relapse Prevention (RP) [12], have demonstrated comparable improvements in both PTSD and substance use outcomes as non-trauma focused integrated treatment [13, 14] and have performed as well as trauma-focused treatments on PTSD and substance use outcomes in *a priori* intent-to-treat models of some [15–18] but not all trials [19]. Recent meta-analytic work specifically compared manualized CBT-SUD control conditions with both trauma-focused and with non-trauma-focused interventions designed for those with PTSD/SUD in the context of randomized clinical trials [20]. The results indicate that the CBT-SUD conditions did not significantly differ from either trauma-focused or non-trauma-focused treatments on PTSD outcomes at either immediate post-test or longest follow-up, but were associated with significantly better substance use outcomes [20]. Additionally, treatment completion for those assigned to individual CBT-SUD control conditions mostly

ranged from 55% to 86% [13, 17–19]; cf. [21, 22], rates that were typically comparable to or better than those found for the target experimental conditions. Thus, there is evidence that standalone RP may be a reasonable option for individuals with co-occurring PTSD/AUD, particularly if they present for care in addiction treatment settings.

There is less information available about standalone PTSD treatments and how they perform for individuals with co-occurring PTSD/AUD or PTSD/SUD. A few studies have provided Prolonged Exposure (PE) to individuals with PTSD/SUD alongside (i.e., not integrated with) substance abuse treatment, such that the treatments were delivered by different clinicians. In a residential setting 60% of those with PTSD/SUD assigned to PE completed it [23]; however, completion rates for PE in outpatient studies involving patients with this comorbidity have been low (39%) [24], very low (3.3%) [25], or not reported [15]. Although treatment retention is substantially higher when PE is integrated with cognitive-behavioral treatment for addiction, as it is in *Concurrent Treatment of PTSD and SUDs using Prolonged Exposure* (COPE) [14, 17], it would be helpful to have alternative standalone treatment options available to address the needs of outpatients with PTSD/SUD.

One potential single-focus intervention for individuals with PTSD/SUD is Cognitive Processing Therapy (CPT), which performs as well as PE for those with PTSD [26]. CPT has some advantages over PE in that it can be provided as a group therapy [27], can be provided in cases where there is not a clear trauma memory [28, 29] such as in cases of early childhood abuse [30] or alcohol-facilitated sexual assaults [31], and can be more acceptable to patients [32] and providers [33]. A retrospective chart review of veterans with PTSD with and without AUD who received CPT found low drop-out for both groups and comparable declines in PTSD (alcohol use was not assessed) [34]. CPT has reduced PTSD and alcohol use frequency relative to waitlist control [35] and was found to be a helpful follow-up treatment after group-based AUD treatment [36]. Despite these promising preliminary findings, CPT has not been compared to an active treatment in the context of an RCT for individuals with comorbid PTSD and AUD.

To address the efficacy of two promising single-focus treatments for those with comorbid PTSD/AUD, this study compared RP and CPT to each other and to an Assessment Only (AO) condition that involved weekly brief telephone safety monitoring and check-ins. Our primary hypotheses were that the active conditions would show greater improvement from baseline to immediate follow-up on PTSD and drinking outcomes than the AO condition. Following re-randomization of AO participants to CPT or RP, both treatments were hypothesized to yield significant within-subject improvements in PTSD and alcohol consumption through the 12-month follow-up period, but those receiving CPT were expected to have better PTSD outcomes and those receiving RP were expected to have better drinking outcomes.

## Materials and methods

The CONSORT Checklist may be found in the online supplemental material (**S1 Table in S1 File**) along with a copy of the IRB-approved protocol.

### Trial design and any changes after trial commencement

This is a parallel group, randomized clinical trial (RCT) wherein participants were randomized to one of two active behavioral interventions or to 6-weeks of assessment-only (AO) with 2:2:1 allocation. Those initially assigned to AO were re-randomized to one of the two active interventions (1:1 allocation) after 6-weeks of assessment. Participants were followed for a maximum of 12-months post-treatment. The trial design remained the same but several changes to methods and procedures were made after the trial commenced to improve recruitment and

participant retention (please see the IRB-approved protocol for an overview of major modifications; major changes are also noted in the methods overview below).

## Participants, eligibility criteria, and settings

The study took place at a university hospital-affiliated mental health clinic and a VA-based research clinic in the Pacific Northwest with participant recruitment taking place from November 13, 2013 and December 8, 2017 and participant contact ending August 31, 2018. One hundred and one adults who responded to recruitment ads in newspapers, flyers, or to letters sent via medical record abstraction and who met current DSM-5 diagnostic criteria for both PTSD and AUD were randomized to one of the three study conditions. Veterans and non-veterans were enrolled at both sites. Additionally, participants needed to meet the following study inclusion criteria: 1) reported at least 2 heavy drinking days (5+/4+ standard drinks for men/women) or at least 2 weeks of heavy drinking (21+/14+ drinks per week for men/women) [37] in the past month {Note: The drinking inclusion criteria were assessed during the initial phone screen and were not re-adjudicated at the subsequent in-person screening assessment. This led to the inclusion of 3 participants (2 initially randomized to CPT and 1 initially randomized to RP) who reported past-30-day abstinence from alcohol at the in-person visit}; 2) stated desire to abstain from or reduce alcohol use; 3) English fluency, and 4) age 18 and older. Study exclusion criteria were: 1) presence of a psychotic or uncontrolled bipolar disorder; 2) suicide attempt or serious self-harm in the past 3 months or suicidal ideation with intent or plan in the past 2 months; 3) current interpersonally violent relationship; 4) alcohol withdrawal at time of consent as assessed by the SHOT (Sweating, Hallucinations, Orientation, and Tremor) [38]; 5) unstable psychiatric medication in the past 3 months (i.e., dose changes and/or addition or subtraction of medications); 6) past-month Antabuse use (Note: Other alcohol medications were allowed (e.g., Naltrexone, Acamprosate, etc.) if participants were on them for at least three months prior to screening because it is possible to drink at unhealthy levels while taking these medication); and 7) past-month trauma-focused mental health treatment or behaviorally focused alcohol treatment. Supportive counseling, self-help, and medication management were allowed during treatment and there were no restrictions on treatment during follow-up.

The study was approved by both the University of Washington and the VA Puget Sound institutional review boards and conducted according to the Declaration of Helsinki and the International Conference on Harmonization's Tripartite Guideline on Good Clinical Practice. All participants underwent a consent process that included a careful review of what the study entailed, addressed any questions or concerns they had, and obtained written informed consent. The trial is registered at clinicaltrials.gov (NCT01663337).

## Interventions and the AO condition

**Cognitive Processing Therapy.** CPT is a 12-session cognitive treatment that teaches clients to question and replace faulty cognitions about causes of traumatic events and overgeneralized beliefs through progressive worksheets and Socratic questioning [39]. Participants were encouraged to experience emotions about their trauma while working on thoughts about it. We used CPT without trauma narratives [39]. Alcohol use in CPT is conceptualized as an avoidance behavior [40] and addressed without protocol modification both through directly targeting it as an avoidance behavior and through challenging alcohol-related cognitive distortions using cognitive therapy.

**Relapse Prevention.** RP is a 12-session CBT approach that emphasizes self-monitoring and coping skills to address high-risk drinking situations [12]. RP works with each clients'

patterns of substance use and can address PTSD-related triggers without protocol modification in that participants were encouraged to use the skills they were learning (e.g., reaching out for social support, engaging in healthy distraction, reminding oneself of reasons for not drinking, etc.) when they experienced PTSD symptoms such as intrusive thoughts or startle. Discussion of trauma memories was proscribed.

**Assessment Only.** AO participants received brief (15 minute) weekly support/safety calls with the study therapist they would see upon re-randomization.

Participants attended individual therapy sessions once or twice per week for a maximum of 20 weeks. The study was originally designed such that participants initially assigned to an active treatment would attend twice weekly sessions over 6-weeks. However, this treatment schedule proved too challenging for most participants and the treatment window was adjusted to allow up to 20 weeks for treatment completion. Thus, the treatment duration during the active treatment phase was 0 to 20 weeks. Of those who completed at least one treatment session, the average number of weeks between first and last session completed was 7.13 ($SD$ = 4.03) for CPT and 7.35 ($SD$ = 3.73) for RP (with AO participants included following re-randomization). Among those who completed at least 9 treatment sessions, the average number of weeks between first and last session completed was 9.65 ($SD$ = 2.71) for CPT and 9.36 ($SD$ = 2.00) for RP.

Three female Licensed Clinical Social Workers learned both treatments via two-day workshops (CPT: Patricia Resick, PhD; RP: Denise Hien, PhD). Two of the three clinicians provided care at the university-based clinic and the third provided care at the VA-based clinic. Intervention fidelity was supported by weekly supervision provided by the study principal investigators (CPT: DK) and (RP: TS). To assess intervention fidelity, trained master's level counselors coded audiotapes of 75 sessions regarding coverage of core treatment elements and intrusions from the other therapy. All CPT sessions and 97% of RP sessions met their respective minimal requirements. Of the sessions coded, two CPT and four RP sessions had intrusions or contaminations from the other treatment. For example, teaching drink refusal skills in the setting of CPT or engaging in Socratic questioning regarding a faulty trauma-related belief in the setting of RP would constitute treatment contaminations.

Consistent with Health and Human Services guidance [41], Adverse Events (AEs) were defined as any abnormal or unanticipated signs or symptoms temporally associated with study participation whether deemed related to study participation. Severe adverse events (SAEs) were defined as death, hospitalization, prolonging of hospitalization, or permanent or persisting disability temporally associated with study participation whether deemed related to study participation. AEs and SAEs were queried during all clinical and research contacts. In addition, for those enrolled at the University of Washington, automatic alerts were provided if a participant visited the Emergency Room or was admitted to the hospital for any reason.

## Outcomes

Assessment interviews were conducted by one of two trained and supervised master's level social workers masked to study condition at baseline, post-treatment, 3- and 12-month follow-up points. Self-report measures collected demographic information (age, gender, race/ethnicity, marital status, employment, and veteran status) at baseline.

The CAPS-5 [42] was used to assess PTSD diagnostic criteria and severity, the latter of which was represented via a total score (possible range: 0 to 80; $\alpha$ = .71 –.94 across assessments). Following Norman et al. [43], remission was defined as total CAPS-5 score < 12. Exposure to 16 potentially traumatic events was assessed via the Life Stressor Checklist [44] at baseline. Number of trauma types and the index trauma were coded.

The Form-90 [45] was administered to ascertain drinking days (DD) and heavy drinking days (HDD). Information from the prior 30 days was used to parallel the PTSD timeframe. Low-risk drinking status was based on NIAAA criteria (i.e., no HDD past 4 weeks and no weeks with $\geq$15/$\geq$8 drinks for men/women) [46]. Past month abstinence status was noted.

## Assessment procedures

Participants were assessed at baseline, immediate post-treatment, and 3- and 12-months post-treatment, with AO participants completing an extra assessment upon treatment completion. (Two additional online assessments took place at 6- and 9-months but because the CAPS and Form-90 interviews were not conducted, these data are not included here.) CPT and RP participants were required to complete the immediate post-treatment assessment within a month of treatment completion. AO participants completed this assessment 6-weeks post baseline to avoid further treatment delay. Because people were allowed 20 weeks in which to complete the 12-session interventions timing from baseline to the first post-treatment assessment was somewhat variable, but the 3- and 12-month assessments took place 3 and 12 months after treatment ended. Maximum remuneration for assessments was $320 for non-AO and $375 for AO participants. To maximize time for enrollment, the final 13 participants were only followed through the 3-month follow-up (or post-treatment assessment for AO participants) rather than out to 12-months as originally planned.

Additionally, participants were asked to complete daily monitoring of symptoms starting the day of the baseline through the end of treatment. These data were used to identify PTSD symptom exacerbations and unsafe alcohol consumption we note that a PTSD exacerbation was characterized by 47+ point elevation over each person's own symptom average during the week prior to starting treatment (possible scores ranged from 0 to 160 as we used a 0–8 point scale to maximize variability across the 20 DSM-5 PTSD symptoms) and a potentially dangerous level of alcohol consumption was characterized by an estimated blood alcohol level of $\geq 0.35$ calculated using baseline weight and number of standard drinks reported via IVR.

## Sample size calculations

Considerable research shows that CPT and RP are effective treatments for their prescribed problems (i.e., PTSD for CPT and HDD for RP) and hence the study was powered to detect effects of each intervention on the non-prescribed targets (i.e., HDD for CPT and PTSD for RP). A priori power analyses were conducted via simulation using mixed effects generalized linear models with different outcome distributions (i.e., normal vs. Poisson). Datasets were generated in which fixed effects for intercept and slope were based on preliminary studies and random-effects were generated based on random draws from a multivariate normal distribution (also specified based on prior research with PTSD and drinking outcomes). The simulation-based estimate of power was provided by the percentage of datasets in which the t statistic for the effect of intervention condition on change across time indicated significance at a critical value of $p < .05$. These procedures indicated a sample of 235 (CPT/RP each 95; AO 45) was needed to have .8 power to detect moderate (d = .4) effect sizes on the non-target outcomes for each active condition relative to the AO condition (i.e., drinking/CPT; PTSD/RP).

## Randomization and allocation concealment

Randomization tables stratified by gender, PTSD severity (CAPS-5 [42] score 35+ vs. below), and AUD severity (MINI [47]; AUD mild/moderate [2–5] vs. severe [6+]) were created prior to participant enrollment by one of the principal investigators (TS) who was not involved with participant allocation. Participants with current PTSD/AUD were initially allocated 2:2:1 to

CPT, RP, or 6-weeks of assessment (AO), after which AO participants completed their first follow-up assessment and were re-randomized to an active condition (1:1 allocation). The Project Coordinator conducted randomization via the pre-set tables. Although allocation was not formally masked, it was not possible to know ahead of time the condition assignment of a particular participant. Because our primary interest was in the comparisons between participants assigned to CPT and to RP and to minimize the number of people whose active treatment receipt would be delayed, we opted to assign fewer participants to the AO condition.

## Masking

Although it is not possible to keep participants or clinicians masked to intervention assignment in behavioral research, when participants were randomized to active treatment they remained masked to CPT/RP assignment until the first therapy session. Additionally, assessors remained masked to condition assignment for all assessments.

## Statistical analyses

Hypotheses were tested via two sets of intent-to-treat mixed effects models. The first set compared CPT and RP to AO with respect to change in PTSD severity, DD, and HDD between baseline and first follow-up. Covariates were included for condition (RP vs. AO, CPT vs. AO), time (baseline = 0; first follow-up = 1), and time-by-condition interaction terms. To increase model precision and reduce bias due to missing data, additional covariates included demographic variables of sex (female = 0, male = 1), age (in years), and race/ethnicity (Hispanic or nonwhite = 0, non-Hispanic White = 1); site at which individuals were treated (university clinic = 0, VA = 1); and whether employed at baseline (unemployed = 0, employed = 1) as a measure of economic status. Although income level also differed between groups, because this was a function of employment status, we opted to covary only employment status. Random effects were included for intercept and time. Primary coefficients of interest were main effects of time, which captured changes from baseline to first follow-up in the AO group, and time-by-condition interactions, which assessed whether CPT and RP differed from AO regarding changes from baseline to follow-up.

The second set of analyses compared CPT and RP across four time points, from baseline through 12-months, including re-randomized AO participants whose first follow-up was treated as their baseline. Condition was treated as a binary factor (0 = CPT, 1 = RP). Change was modeled with a piecewise, or spline, function of time. To assess overall changes from pre- to posttreatment, the first time variable reflected baseline (0) vs. posttreatment (1 = immediate posttreatment, 1 = 3-month follow-up, 1 = 12-month follow-up). The second time coding captured change (in units of months) during the year following treatment (0 = pretreatment; 0 = immediate posttreatment; 3 = 3-month follow-up; 12 = 12-month follow-up). Interaction terms were included between condition and each coding of time. This approach addressed (1) whether participants had immediate improvements after treatment (main effect of pre-post time), (2) whether these improvements differed between CPT and RP (pre-post time-by-condition interaction), (3) whether improvements increased or decreased during the year after treatment (main effect of post-treatment time), and (4) whether change during the year post-treatment differed between CPT and RP (post-treatment time-by-condition interaction). AO assignment was added as a covariate to the original sociodemographic covariates.

Linear forms of the mixed model were used for PTSD severity. For PTSD severity, effect sizes are reported in terms of Cohen's *d*, which were computed as the time-by-treatment interaction estimate divided by the pooled standard deviation for PTSD symptom severity at baseline. Over-dispersed Poisson models were used for DD and HDD (discrete non-negative

integers showing positive skew; i.e., count outcomes). The over-dispersed Poisson model includes an extra term to account for the variance of outcomes exceeding the means [48, 49], which was the case for the DD and HDD outcomes. For these models, effect sizes are reported as Count Ratios (CRs), which describe proportional change in the count associated with a 1-unit increase in the covariate of interest (e.g., the treatment group relative to the reference) [50, 51]. Between-group differences in PTSD remission, alcohol abstinence, and low-risk drinking were evaluated via Chi-square, exact tests. All other models were estimated with HLM 7 using restricted maximum likelihood (REML) estimation. REML is recommended for growth models with small to moderate sample sizes and in situations where all cases are not assessed at all time points. The assumption of normally distributed random effects was checked for all models by saving random effect estimates and then examining their distributions graphically and with respect to measures of skewness and kurtosis. As an approach to handling missing data, simulation studies indicate REML performs as well as multiple imputation [52]. The models used all available data, including data from individuals who were not assessed at all time points. This approach provides unbiased estimates when data are missing at random (MAR: i.e., missingness related only to measured covariates) [53, 54]. To assess sensitivity to violations of this MAR assumption, we evaluated pattern mixture models [55] appropriate for data missing not at random (MNAR) to investigate whether treatment effects varied as a function of attrition. Specifically, we created a dummy variable for attrition, defined as missing at least one follow-up assessment ($n = 55$; coded as 1), relative to completing all follow-up assessments ($n = 46$; coded as 0). Then, within the four-timepoint models for each outcome, we examined a three-way interaction between attrition, treatment, and each time variable, as well as lower-order effects (i.e., two-way interactions and main effects). Wald tests revealed that inclusion of main and interactive effects of attrition did not significantly improve model fit for any outcome, consistent with the MAR assumption of the primary analyses presented below.

## Results

### Participant flow (recruitment, treatment, and assessment completion)

Recruitment, treatment, and assessment completion information is in the CONSORT diagram (shown in **Fig 1**). Briefly, 826 individuals completed the study phone screen and 262 completed the in-person eligibility interview. Of these, 101 were randomized to one of the three study conditions. Treatment completion, defined as attending 9–12 sessions, was similar for CPT (53%) and RP (58%) {note: proportions include re-randomized AO participants, but treatment completion proportions were similar for those initially assigned to active or delayed (AO) care}. With regard to assessment completion, AO participants were more likely to complete the first follow-up (96%) than active treatment participants (CPT: 68%, RP: 74%; $\chi^2(2) = 6.05$, $p = .049$). The number of weeks between baseline and post-treatment assessment were, on average, 9.37 ($SD = 2.49$) for the AO group, 13.48 ($SD = 4.16$) for the CPT group, and 14.33 ($SD = 3.36$) for the RP group. There were no statistically significant differences in assessment completion for CPT and RP.

### Participant characteristics

Participants' mean age was 42.10 ($SD = 13.07$) and 56% were female. Most identified as White race (53%), followed by Black (18%), multiracial (13%), Asian (5%), American Indian/Alaskan Native (5%), and other (6%); 21% indicated Hispanic ethnicity of any race. Most were single/never married (60%), 33% were college graduates, 29% were veterans, and 38% were employed. Participants reported their worst traumatic events as sexual assault (36%), physical

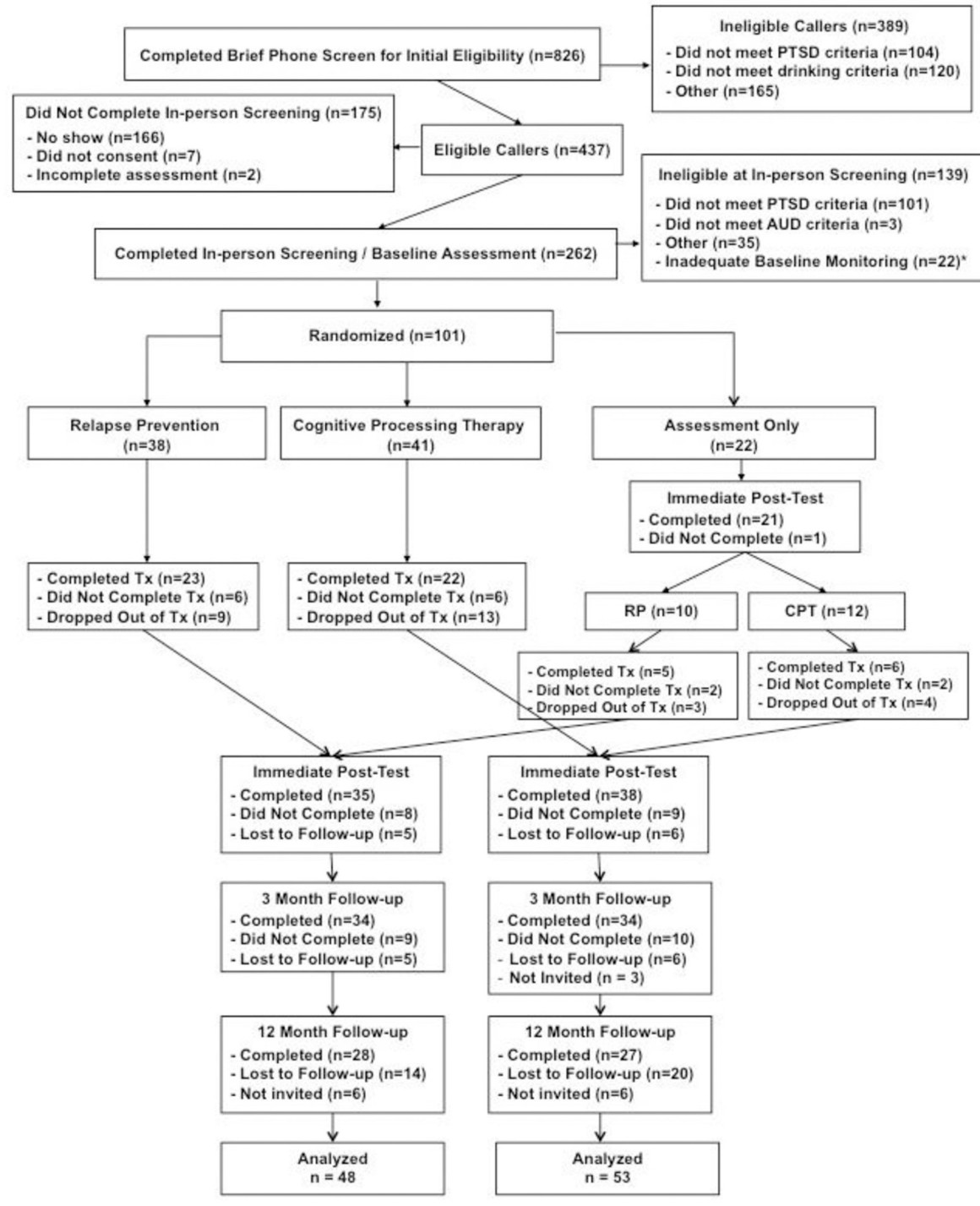

**Fig 1. CONSORT diagram.**

assault (33%), traumatic loss (11%), combat (3%), or another trauma type (18%). Other current substance use included cannabis (57%), tobacco (45%), and other drugs (22%); 39% met criteria for a non-tobacco Drug Use Disorder as assessed via the MINI adapted for DMS-5 [47]. Detailed sociodemographic information is provided by condition in the online supplement (**S2 Table in S1 File**), as is baseline clinical information (**S3 Table in S1 File**). Sociodemographic variables did not differ by condition ($p > .05$) except that RP participants were more likely to be employed and relatedly, less likely to be impoverished relative to those assigned to the other conditions and they were younger than CPT participants. Clinical characteristics did not differ by group with following exceptions: 1) those assigned to CPT were significantly more likely to report that their first alcohol problem(s) preceded their index trauma than were those assigned to RP ($p = .039$), and 2) those assigned to CPT were also significantly older than those assigned to RP when their index trauma occurred ($p = .021$).

## CPT and RP compared to AO

Table 1 (Panel A) displays outcome means by initial condition assignment at baseline and first follow-up. There was a significant site effect with participants enrolled at the VA drinking less at baseline and follow-up (DD: $p = .013$; HDD: $p = .009$) (Table 2). AO participants showed statistically significant improvements in PTSD severity ($p = .014$) and HDD ($p = .013$), but not DD ($p = .408$), as reflected by the main effect of time. The effect of CPT vs. AO (time-by-CPT interaction) on reducing PTSD symptom severity was statistically significant ($d = 1.22$, $p = .004$). The effect of RP vs. AO (time-by-RP interaction) on reducing PTSD was not statistically significant ($d = 0.76$, $p = .070$) and nor was the effect of CPT vs RP ($d = 0.46$, $p = .234$), which favored the CPT condition. Time-by-condition terms were not statistically significant for DD (CPT vs AO: $p = .171$; RP vs AO: $p = .145$), but they were for HDD such that CPT and RP participants decreased their HDD 49% ($p = .015$) and 66% ($p < .001$) more, respectively, than did AO participants.

**Table 1. Descriptive statistics for main outcomes stratified by condition.**

Panel A. *Baseline and follow-up by original assignment.*

| | Baseline | | | | | | Immediate Post-treatment | | | | | | | | | |
| --- | --- | --- | --- | --- | --- | --- | --- | --- | --- | --- | --- | --- | --- | --- | --- | --- |
| | AO (n = 22) | | CPT (n = 41) | | RP (n = 38) | | AO (n = 21/18)[a] | | CPT (n = 28/25)[a] | | RP (n = 28/26)[a] | | | | | |
| | M | SD | M | SD | M | SD | M | SD | M | SD | M | SD | | | | |
| PTSD symptom severity | 32.09 | 8.12 | 33.51 | 7.64 | 34.11 | 8.43 | 25.90 | 11.32 | 17.43 | 11.26 | 21.50 | 12.36 | | | | |
| Drinking days in past 30 d | 18.64 | 9.86 | 16.71 | 10.73 | 18.61 | 8.72 | 15.00 | 11.03 | 8.72 | 11.36 | 9.54 | 9.53 | | | | |
| Heavy drinking days in past 30 d | 14.59 | 10.67 | 12.78 | 10.81 | 13.27 | 6.68 | 9.06 | 10.76 | 3.40 | 5.50 | 2.38 | 3.45 | | | | |

Panel B. *Pre-treatment to 12-month follow-up by treatment assignment.*

| | Pre-treatment | | | | Post-treatment | | | | 3-month follow-up | | | | 12-month follow-up | | | |
| --- | --- | --- | --- | --- | --- | --- | --- | --- | --- | --- | --- | --- | --- | --- | --- | --- |
| | CPT (n = 53/52)[a] | | RP (n = 47/45)[a,b] | | CPT (n = 38/34)[a] | | RP (n = 35/33)[a] | | CPT (n = 34) | | RP (n = 34) | | CPT (n = 27) | | RP (n = 28) | |
| | M | SD | M | SD | M | SD | M | SD | M | SD | M | SD | M | SD | M | SD |
| PTSD symptom severity | 32.28 | 9.08 | 32.00 | 9.62 | 19.47 | 14.01 | 20.09 | 11.96 | 20.38 | 13.41 | 19.71 | 12.63 | 19.69 | 13.27 | 19.79 | 11.72 |
| Drinking days in past 30 d | 16.62 | 10.36 | 17.73 | 9.72 | 9.35 | 11.61 | 10.03 | 10.13 | 8.29 | 9.74 | 8.44 | 10.01 | 8.96 | 10.59 | 8.96 | 11.14 |
| Heavy drinking days in past 30 d | 11.92 | 10.62 | 12.68 | 10.21 | 4.94 | 8.25 | 2.97 | 5.62 | 4.88 | 8.44 | 3.85 | 6.87 | 4.63 | 7.41 | 3.36 | 6.09 |

Note. AO = assessment only, CPT = cognitive processing therapy, RP = relapse prevention, PTSD = Post-traumatic stress disorder. [a]Some cases had missing data on drinking days and heavy drinking days at the second/post-treatment data collection time point; numbers to the left of the forward slash reflect the n*s* for PTSD severity while n*s* to the right are for drinking days and heavy drinking days. [b]One case whose original assignment was to AO and then was assigned to RP did not complete the first follow-up assessment and was thus missing data at the pre-treatment time point.

**Table 2. Model estimates for initial assignment.**

| | PTSD Severity | | Drinking Days | | | Heavy Drinking Days | | |
|---|---|---|---|---|---|---|---|---|
| | *b* | 95% CI | *b* | CR | 95% CI of CR | *b* | CR | 95% CI of CR |
| *Covariates* | | | | | | | | |
| Sex (female = 0, male = 1) | 2.22 | −0.99, 5.42 | 0.13 | 1.14 | 0.89, 1.46 | 0.18 | 1.20 | 0.86, 1.68 |
| Age (years) | −0.14* | −0.28, −0.01 | 0.00 | 1.00 | 0.99, 1.01 | 0.00 | 1.00 | 0.99, 1.02 |
| Race/ethnicity (Hispanic or nonwhite = 0, non-Hispanic white = 1) | 1.83 | −1.27, 4.93 | 0.11 | 1.11 | 0.88, 1.41 | 0.25 | 1.28 | 0.93, 1.77 |
| Site (University = 0, VA = 1) | −0.90 | −4.12, 2.32 | −0.32* | 0.73 | 0.57, 0.93 | −0.45** | 0.64 | 0.45, 0.89 |
| Employed (not employed = 0, employed = 1) | −3.83* | −7.47, −0.19 | −0.03 | 0.97 | 0.74, 1.28 | −0.57** | 0.57 | 0.39, 0.84 |
| *Main effects* | | | | | | | | |
| Time (baseline = 0, follow-up = 1) | −6.29* | −11.22, −1.36 | −0.17 | 0.85 | 0.57, 1.26 | −0.46* | 0.63 | 0.44, 0.91 |
| CPT (vs. AO) | 1.98 | −2.11, 6.08 | −0.11 | 0.90 | 0.67, 1.23 | −0.04 | 0.96 | 0.64, 1.44 |
| RP (vs. AO) | 2.06 | −2.29, 6.42 | −0.07 | 0.93 | 0.68, 1.29 | 0.03 | 1.03 | 0.66, 1.59 |
| *Time x condition interactions* | | | | | | | | |
| Time x CPT (vs. AO) | −9.72** | −16.20, −3.23 | −0.38 | 0.68 | 0.39, 1.18 | −0.67* | 0.51 | 0.30, 0.88 |
| Time x RP (vs. AO) | −6.07 | −12.56, 0.42 | −0.39 | 0.68 | 0.40, 1.15 | −1.09*** | 0.34 | 0.19, 0.59 |

*Note.* Fixed effects estimates for models of cognitive processing therapy (CPT) and relapse prevention (RP) compared to assessment only (AO) across two time points (baseline to first follow-up). AO = assessment only, CPT = cognitive processing therapy, RP = relapse prevention, PTSD = post-traumatic stress disorder, VA = Veteran's Administration, *CR* = count ratio. ***p < .001, **p < .01, *p < .05.

Panel A of **S4 Table in S1 File** shows the proportion across conditions meeting criteria for PTSD remission, abstinence, and low-risk drinking at baseline and immediate follow-up. Although more than twice as many active condition participants showed clinically meaningful change on these indices at post-treatment than AO participants, the only significant between-group difference detected was in PTSD remission between AO and CPT (exact test *p* = .040).

## CPT versus RP

Table 1 (Panel B) shows means of PTSD and drinking outcomes by RP vs. CPT at four time points with the original AO participants re-randomized to one of the active treatment arms. **Fig 2** displays model-predicted marginal means by condition and Table 3 shows model estimates of all fixed effects. Change within CPT is reflected in significant, negative main effects of pre-post time (PTSD: *p* < .001; DD: *p* = .004; HDD: *p* < .001). Interactions between condition and pre-post time reflect that decreases in PTSD severity were slightly greater for individuals in the CPT condition compared to those in the RP condition (*d* = 0.14, *p* = .651) while decreases in DD were slightly greater for individuals in the RP condition (*CR* = 0.85, *p* = .493), but neither interaction was statistically significant. The pre-post condition-by-time interaction was significant for HDD (*p* = .037): RP participants showed a 45% greater reduction in HDD than did CPT participants. All main effects of monthly change across the post-treatment period were close to zero and non-significant (PTSD: *p* = .385; DD: *p* = .912; HDD: *p* = .680), reflecting stability in outcomes after treatment. Overall, participants in CPT and RP showed substantial initial decreases in PTSD (CPT: 47%; RP: 35%) and heavy drinking (CPT: 73%; RP: 82%), and these decreases were maintained for both outcomes across both conditions throughout the follow-up period.

Panel B of **S4 Table in S1 File** shows proportions of participants by condition achieving PTSD remission, abstinence, and low-risk drinking over time. Across follow-ups, PTSD remission rates were 23% to 41%; low-risk drinking rates were 42% to 52%; abstinence rates were 28% to 41%; rates of combined PTSD remission and low-risk drinking ranged from 15% to

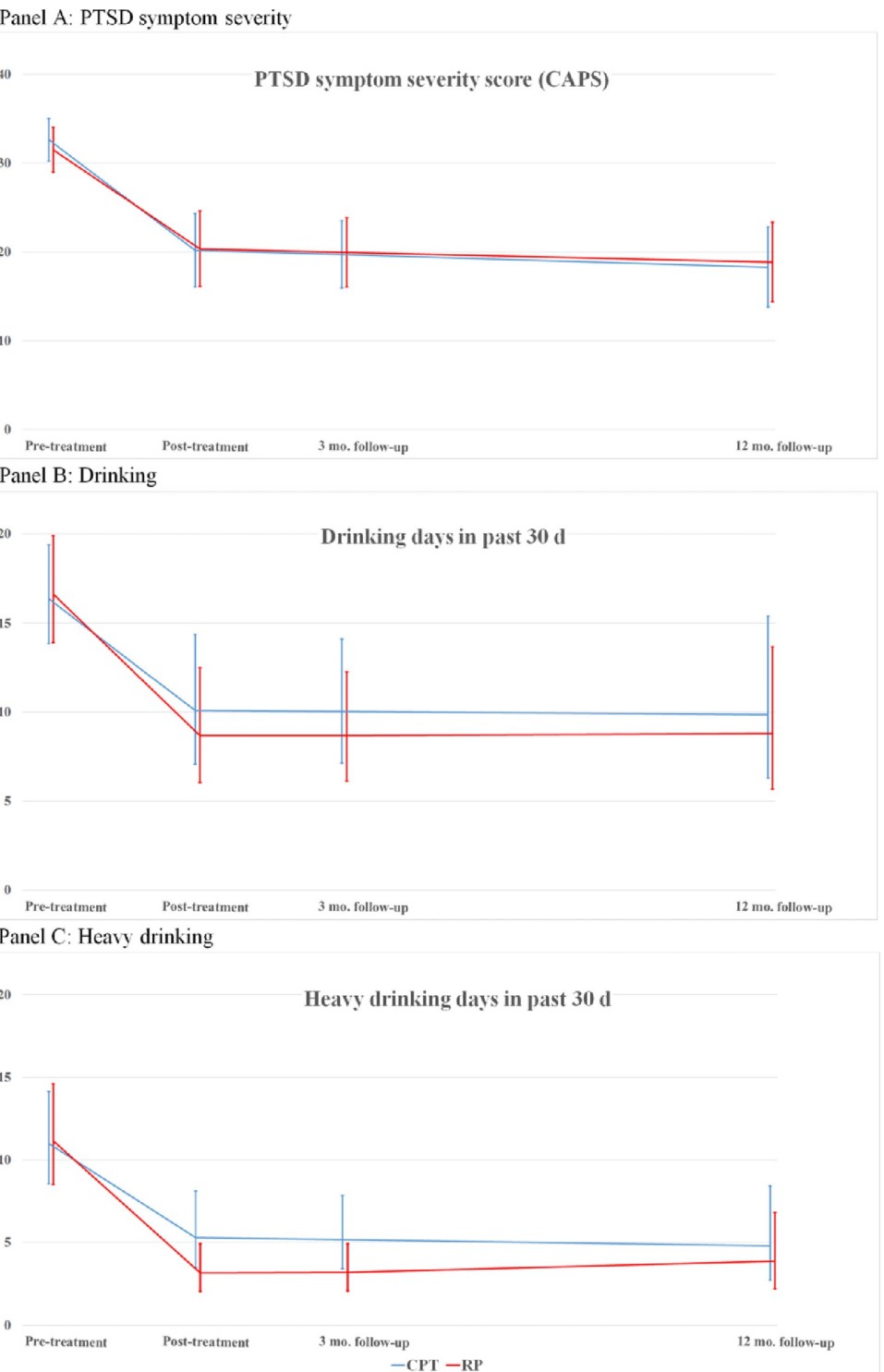

**Fig 2.** Model predicted means and 95% confidence intervals for PTSD symptom severity (Panel A), drinking days in the past 30 (Panel B), and heavy drinking days in the past 30 (Panel C) by treatment condition from pre-treatment to 12-month follow-up.

**Table 3. Model estimates for final treatment assignment.**

| | PTSD Severity | | Drinking Days | | | Heavy Drinking Days | | |
|---|---|---|---|---|---|---|---|---|
| | *b* | 95% CI | *b* | CR | 95% CI of CR | *b* | CR | 95% CI of CR |
| *Covariates* | | | | | | | | |
| Sex (female = 0, male = 1) | 3.09 | −0.47, 6.66 | 0.16 | 1.17 | 0.90, 1.53 | 0.19 | 1.22 | 0.82, 1.79 |
| Age (years) | −0.14 | −0.30, 0.01 | 0.01 | 1.01 | 1.00, 1.02 | 0.01 | 1.01 | 0.99, 1.02 |
| Race/ethnicity (Hispanic or nonwhite = 0, non-Hispanic white = 1) | 1.83 | −1.61, 5.28 | 0.12 | 1.13 | 0.88, 1.45 | 0.43* | 1.53 | 1.06, 2.23 |
| Site (University = 0, VA = 1) | −0.25 | −3.86, 3.35 | −0.38** | 0.69 | 0.52, 0.90 | −0.47* | 0.63 | 0.42, 0.93 |
| AO (initial assignment CPT or RP = 0; initial assignment AO = 1) | −6.89** | −10.99, −2.79 | −0.16 | 0.85 | 0.62, 1.17 | −0.54* | 0.58 | 0.36, 0.93 |
| Employed (not employed = 0, employed = 1) | −3.44 | −7.48, 0.61 | −0.04 | 0.96 | 0.72, 1.29 | −0.60** | 0.55 | 0.35, 0.86 |
| *Main effects* | | | | | | | | |
| RP (CPT = 0, RP = 1) | −1.13 | −4.70, 2.44 | 0.01 | 1.02 | 0.79, 1.31 | 0.01 | 1.01 | 0.69, 1.49 |
| Pre-post time (pretreatment = 0, posttreatment and 3- and 12-month follow-up = 1) | −12.45*** | −16.32, −8.57 | −0.49** | 0.61 | 0.44, 0.85 | −0.73*** | 0.48 | 0.33, 0.70 |
| Post-treatment time (pretreatment and posttreatment = 0, 3-month = 3, 12-month = 12) | −0.16 | −0.51, 0.20 | 0.00 | 1.00 | 0.97, 1.03 | −0.01 | 0.99 | 0.95, 1.03 |
| *Time x condition interactions* | | | | | | | | |
| Pre-post time x RP | 1.29 | −4.29, 6.87 | −0.16 | 0.85 | 0.53, 1.36 | −0.59* | 0.55 | 0.32, 0.97 |
| Posttreatment time x RP | 0.03 | −0.47, 0.53 | 0.00 | 1.00 | 0.96, 1.05 | 0.03 | 1.03 | 0.97, 1.09 |

*Note.* Estimates for models of the effects of relapse prevention (RP) compared to cognitive processing therapy (CPT) across four time points (pre-treatment through 12-month follow-up). AO = assessment only, CPT = cognitive processing therapy, RP = relapse prevention, PTSD = post-traumatic stress disorder, VA = Veteran's Administration, *CR* = count ratio.

***p < .001

**p < .01

*p < .05.

21%, and rates of combined PTSD remission and abstinence ranged from 6% to 18%. There were no significant between-group differences on any of the remission indicators.

## Completers

Both two and four time-point analyses involving only treatment completers (i.e., those who completed at least 9 treatment sessions; *n* = 56; 28 of 53 assigned to CPT and 28 of 48 assigned to RP) revealed a similar pattern of results as in primary analyses (see online supplement **S5-S7 Tables in S1 File**).

## Safety

Approximately three-quarters of participants experienced at least one of the following: PTSD exacerbation, alcohol consumption in excess of .35 BAC, suicidal ideation at some point during the study. Because none of these situations were unanticipated for this clinical group, they were not considered AEs by either IRB or the study funder. There were, however, 19 SAEs involving 15 participants, all of which occurred after treatment ended. One of the SAEs was likely study related; a participant was hospitalized for psychosis three days after his final CPT session during which he noted that treatment opened up new feelings. There was one death (CPT) due to a long-standing medical issue, 8 psychiatric hospitalizations (7 CPT, 1 RP), and 10 medical hospitalizations (4 CPT, 6 RP), 5 of which were likely for substance-related health

issues (3 CPT, 2 RP). The psychiatric hospitalizations for those assigned to CPT were for the following concerns: 1 for psychosis, 3 for mood disorders, 1 for stress, and 2 for suicidal and/or homicidal ideation. The psychiatric hospitalization for the person assigned to RP was for suicidal ideation. There were no known suicide deaths or attempts.

## Discussion

CPT and RP were delivered unmodified to evaluate whether these single-focus treatments for PTSD and AUD, respectively, lead to sustained improvements on both types of outcomes pertinent to individuals with PTSD/AUD. CPT addressed avoidance functions of drinking and thoughts hindering PTSD recovery (e.g., "I can't cope with my rape memories without drinking"). Although explicit discussion of trauma memories was prohibited, RP addressed PTSD symptoms that triggered craving or drinking by encouraging use of standard behavioral coping strategies such as seeking social support, engaging in healthy distraction or alternative activities, waiting for cravings to pass, and reminding oneself of one's reasons for not drinking.

Although caution is warranted as our models were underpowered, our primary hypotheses were largely supported. Specifically, compared to AO participants, those receiving CPT showed significantly greater reduction in PTSD severity that was associated with a large effect size ($d$ = 1.22). In contrast, although the effect size pertaining to PTSD for the RP v. AO comparison was moderately large ($d$ = .76), it failed to reach statistical significance. However, both active conditions were associated with significantly greater reductions in heavy drinking days immediately post-treatment than assignment to the AO condition, differences that were substantial and clinically meaningful (CPT v. AO: 49%; RP v. AO: 66%). Although receipt of an active intervention was more beneficial than assignment to AO, it is noteworthy that those assigned to the AO condition made significant within-subject improvements in both their PTSD symptoms and their HDD. These findings are consistent with other research showing that many people reduce their drinking after making the decision to seek care but prior to actual receipt of care [56], and that many people with PTSD benefit from non-directive phone support [57].

As predicted, after AO participants were re-randomized we found that both active treatments were associated with substantial within-person improvement on both PTSD severity and drinking outcomes. Participants in both active conditions showed robust initial decreases in PTSD severity as measured by CAPS scores from pre- to post-treatment (CPT: 48%; RP: 37%) and in days of heavy drinking (CPT: 73%; RP: 82%). Of note, the pattern of results suggests that both active treatments were more associated with harm reduction oriented changes in drinking (i.e., reductions in heavy drinking) than they were with abstinence. With regard to between-group comparisons, we found that the two groups did not differ significantly on either PTSD severity or days drinking, but as predicted, RP participants showed significantly greater reductions in heavy drinking days than those receiving CPT (45% greater reduction in heavy drinking for RP v CPT).

Overall, the percent reduction in both PTSD and drinking outcomes from baseline to post-treatment for both active conditions tested in the present study was comparable to those found among individuals assigned to the integrated intervention COPE in trials addressing comorbid PTSD/SUD, which for PTSD range from 26% to 40% (current study: CPT 40%; RP: 37%) and for primary substance use outcomes range from 36% to 65% (current study: CPT 59%; RP: 77%) [16, 19, 43, 58]. In the present study, CPT and RP participants were nearly identical in achievement of "low-risk drinking" status and past-month abstinence at 12-months. Approximately one-fifth of both CPT and RP participants achieved both PTSD remission and low-risk drinking status at 12-months, though fewer achieved both PTSD remission and abstinence.

When considered alongside the extant findings in the literature pertaining to COPE, the present findings have implications for service delivery for PTSD/AUD patients in that they suggest that unmodified CPT and unmodified RP may constitute additional viable treatment options when COPE, or other effective integrated treatment options, are not available [18].

Applying CPT and RP effectively in the setting of PTSD/AUD likely requires sustained attention to each treatments' non-target issue (i.e., drinking/CPT, PTSD/RP) and skillful application of each treatments' strategies to address patients' full clinical presentations. In the present study, regular supervision addressed patients' comorbid presentations within each treatment frame. Additionally, treatment was informed by intensive symptom monitoring, an uncommon practice in general mental health or addiction clinics. However, in this controlled therapeutic context, results suggest that CPT and RP have potential to treat co-occurring disorders as the magnitude of change out to 12-months in both PTSD symptom severity and substance use observed in the current study is generally comparable to that seen for COPE in trials with shorter follow-up periods [16, 19, 43, 58].

Our findings also raise important considerations regarding patient safety. There were no unanticipated AE's and no SAE's occurred in either intervention arm during active treatment. However, 15 participants in our study experienced SAE's during follow-up. Both the extant literature and our own results suggest that CBT-SUD interventions, including RP, are safe and well tolerated [13, 15–17]. The present findings regarding the safety of CPT are, however, less clear. Few other studies have examined CPT without SUD intervention strategies integrated in the treatment of PTSD/SUD patients [34, 35] and scant relevant information on safety is available. In a study of culturally adapted CPT tested within a sample of Native American women with PTSD/AUD, there were no AE's or SAE's [35] (personal communication) although inclusion criteria included subthreshold PTSD and some heavy episodic drinking in the past year. A recent pilot study by Vujanovic et al. [59] (personal communication, August 2020) testing an integrated version of CPT and CBT for SUD had one psychiatric SAE among those randomized to the experimental treatment. More research is needed to evaluate CPT delivered to those with more severe AUD, including over longer follow-up periods to evaluate safety post-treatment, and to help explain the discrepancy between this study's findings and prior research on CPT with substance use comorbidity. However, based on these preliminary findings, results suggest that when delivering standalone CPT to those with fairly severe AUD, patients' mood and stress levels should be monitored, including after the primary course of treatment is over, perhaps through tapered clinical contacts or variable length CPT to ensure a good end state[60, 61].

An important next step in this line of research will be determining which patients with comorbid PTSD/AUD are apt to fare well with regard to their PTSD and alcohol outcomes and with regard to safety in solidly executed RP or CPT, and which are best served by integrated interventions such as COPE. Project HARMONY [8] is a recently completed person-level meta-analytic study that is using relevant trial data to identify patient characteristics associated with better (and worse) outcomes reported by participants who received different interventions. In the meantime, clinicians may consider employing shared decision-making in treatment selection when multiple empirically well supported treatments are available [62–67].

The current study has several limitations. Recruitment was challenging, which led to a smaller than anticipated sample size. In many ways this difficulty with recruitment makes sense because patients may have preferences for treatment of their substance use or their PTSD symptoms and may not them as equally desirable treatment options. Whatever the reasons for the recruitment difficulties, our final sample size was 101, which, although comparable to or larger than sample sizes used in most efficacy trials of psychosocial treatments for comorbid PTSD and AUD or SUD [68], does mean we had limited statistical power and precision for estimates of what might be small-to-medium (e.g., $d < 0.4$) effects. We did find that

relative to assessment only there were moderate to large effect sizes favoring the active treatment interventions on both PTSD and heavy drinking. Additionally, the lack of noteworthy difference observed between the two active treatment conditions with regard to PTSD suggests that a non-inferiority trial with a large sample would be helpful in establishing whether individuals with PTSD/AUD can successfully address their PTSD symptoms through receipt of either CPT or RP and could also address mechanisms of change to explore how it is that such different approaches result in similar outcomes. It is possible that general behavioral skills clients gain through both CPT and RP (e.g., improved communication and assertiveness, distress tolerance, reaching out for healthy social support, coping self-efficacy) are helpful in addressing underlying dimensions of psychopathology common to both SUD and PTSD. Gaining such skills may help clients mitigate current stress, better tolerate negative emotions, and temper stress reactivity [69–72], including those associated with cravings in response to substance or trauma-related cues. It is also possible that improvement in one or the other set of signs and symptoms (i.e., PTSD or AUD) leads to salutary effects in the other. Indeed, support for the idea that reductions in PTSD are associated with later reductions in substance use was found [73] in a large behavioral treatment trial involving women with co-occurring PTSD and SUD. This secondary analysis did not, however, find that reductions in substance use were associated with later improvements in PTSD. Future research on both the sequence of symptom change and mechanisms of change associated with these two interventions will be important for unraveling the finding that they are each associated not only with positive changes in their target outcomes, but also in their non-target outcomes.

The lack of 12-month assessment for the final 13 participants due to time and budget constraints is a limitation as are the relatively high treatment and assessment dropout rates. An appropriate missing data strategy was used to include all cases in the analyses (i.e., including individuals who did not complete assessments at some time points), and sensitivity analyses with pattern mixture models suggested findings were robust to the violation of the assumption that data were missing at random, but missing data do reduce the precision of model estimates. Additionally, the project coordinator conducted phone screens and handled the randomization and thus allocation to condition was not masked. Finally, because active treatment lasted 0 to 20 weeks, assessment timing was variable and it is possible that those originally assigned to assessment only might have continued to improve with regard to their PTSD and drinking if they waited another four to five weeks before being assessed.

The current study also has several noteworthy strengths. This study is one of a small number of RCTs on this topic have included 12-month follow-up assessments [36, 74]. In addition, we recruited a severe sample in that all participants met full diagnostic criteria for current PTSD and AUD and had engaged in recent unsafe drinking. Relatedly, nearly 40% had a concomitant, current DUD. The study design also included a minimal support condition from which people were re-randomized to active treatment in order to gauge the amount of short-term improvement that may be expected after people make a decision to seek care, which in the current study was fairly substantial but, as predicted, did not match the degree of change reported by those in the active treatment conditions. We also benchmarked drinking outcomes against the NIAAA guidelines for safer drinking, which, to our knowledge, is novel in this literature. Finally, we provided estimates regarding rates of clinically meaningful change on both PTSD and drinking outcomes and their combination.

## Conclusions

This randomized clinical trial found that unmodified CPT and RP led to substantial improvements relative to assessment only and, importantly, that following re-randomization of the

control group to active treatment, the two interventions showed substantial comparable improvements in PTSD over the course of 12 months. Both interventions had treatment retention rates and led to improvements that are comparable to those observed for integrated trauma-focused/SUD treatments in other PTSD/SUD trials. Results from this trial suggest that treatments targeting one *or* the other aspect of the PTSD/AUD comorbidity may have salutary effects on both types of outcomes. If these findings are replicated in a future, fully powered trial that mitigates potential safety issues among CPT clients with more severe AUD, it would suggest that mental health and addiction clinics could offer high quality CPT or RP to their patients with comorbid PTSD/AUD as an additional option to integrated interventions such as COPE [7, 8] or as an alternative in settings where access to those integrated interventions is limited.

## Supporting information

**S1 File.**
(DOCX)

**S1 Protocol.**
(DOC)

## Acknowledgments

We very much appreciate the contributions of our colleagues Dr. G. Alan Marlatt, Dr. Scott Coffey, Dr. David Atkins, Dr. Mary Larimer, Dr. Kristen Lindgren, Marti Hickey, Chrys Potuzak, Jordan Royal, Jessica Kicha-Hu, Cameron Paine-Thaler, Kim Hodge, and Lisa Batten. Contributors have given their written permission to be so named with the exceptions of Drs. Marlatt and Coffey, both of whom unfortunately pre-deceased completion of the trial. We also appreciate the time and effort our participants devoted to this study. NIAAA conducted peer review on the study design and oversaw conduct of the study; collection, management, analysis. NIAAA supported preparation of the manuscript but was not involved in interpretation of the data, review, or approval of the manuscript. NIAAA also did not have a role in deciding whether to submit the manuscript for publication.

Finally, we acknowledge that this research was conducted on the traditional, unceded ancestral lands of the Coast Salish people, specifically the Duwamish, Muckleshoot, Stillaguamish, and Suquamish Tribes. The Coast Salish people have always been here, are still here, and will always be here, continuing to honor and bring light to their ancient heritage. We honor with gratitude the land itself and the Duwamish, Muckleshoot, Stillaguamish, and Suquamish people, past and present.

### Ethics statement

This study protocol was reviewed and approved by the Human Subjects Institutional Review Boards of the University of Washington (approved protocol number 39884) and of the VA Puget Sound Healthcare System (approved protocol number 00488). Written informed consent was obtained from all participants.

## Author Contributions

**Conceptualization:** Tracy L. Simpson, Debra L. Kaysen, Denise A. Hien, Lucy Berliner, Patricia A. Resick.

**Data curation:** Charles B. Fleming, Anna E. Jaffe, Sruti Desai.

**Formal analysis:** Charles B. Fleming, Isaac C. Rhew, Anna E. Jaffe.

**Funding acquisition:** Tracy L. Simpson, Debra L. Kaysen.

**Investigation:** Tracy L. Simpson, Debra L. Kaysen.

**Methodology:** Tracy L. Simpson, Debra L. Kaysen, Sruti Desai, Denise A. Hien, Patricia A. Resick.

**Project administration:** Tracy L. Simpson, Debra L. Kaysen, Sruti Desai.

**Resources:** Tracy L. Simpson, Debra L. Kaysen.

**Supervision:** Tracy L. Simpson, Debra L. Kaysen, Sruti Desai, Denise A. Hien, Lucy Berliner, Patricia A. Resick.

**Writing – original draft:** Tracy L. Simpson, Debra L. Kaysen, Charles B. Fleming, Isaac C. Rhew, Anna E. Jaffe.

**Writing – review & editing:** Tracy L. Simpson, Debra L. Kaysen, Charles B. Fleming, Isaac C. Rhew, Anna E. Jaffe, Denise A. Hien, Lucy Berliner, Dennis Donovan, Patricia A. Resick.

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
