## [Decision Letter · Decision Letter 0]

28 Mar 2022

PONE-D-22-03482Cognitive Processing Therapy or Relapse Prevention for Comorbid Posttraumatic Stress Disorder and Alcohol Use Disorder: A Randomized Clinical TrialPLOS ONE

Dear Dr. Simpson,

Thank you for submitting your manuscript to PLOS ONE. After careful consideration, we feel that it has merit but does not fully meet PLOS ONE’s publication criteria as it currently stands. Therefore, we invite you to submit a revised version of the manuscript that addresses the points raised during the review process.

We look forward to receiving your revised manuscript.

Kind regards,

Bernard Le Foll, M.D., Ph.D.

Academic Editor

PLOS ONE

Journal Requirements:

"Dr. Kaysen is a co-author on a book on Cognitive Processing Therapy published by Elsevier for which she receives royalties. In addition she has conducted clinical workshops on Cognitive Processing Therapy for which she has received speakers fees, which could constitute a conflict of interest. Dr. Resick is a co-author on the Cognitive Processing Therapy treatment manual for which she receives royalties and she conducts clinical workshops on Cognitive Processing Therapy for which she receives speakers fees, which could constitute a conflict of interest. The other co-authors have no conflicts of interest to declare pertinent to this submission."

Additional Editor Comments:

This is an interesting manuscript. Comments from the reviewers should be addressed

Reviewers' comments:

Reviewer's Responses to Questions

**Comments to the Author**

1. Is the manuscript technically sound, and do the data support the conclusions?

Reviewer #1: Yes

Reviewer #2: Partly

Reviewer #3: Yes

2. Has the statistical analysis been performed appropriately and rigorously? 

Reviewer #1: Yes

Reviewer #2: Yes

Reviewer #3: Yes

3. Have the authors made all data underlying the findings in their manuscript fully available?

Reviewer #1: No

Reviewer #2: Yes

Reviewer #3: Yes

4. Is the manuscript presented in an intelligible fashion and written in standard English?

Reviewer #1: Yes

Reviewer #2: Yes

Reviewer #3: Yes

5. Review Comments to the Author

Reviewer #1: The objective of this RCT is to assess and compare the CPT treatment, an AUD treatment, and an assessment-only (AO) treatment, via a longitudibal study design. The study was approved by the respective IRB/Ethics Committee, and carries a valid NCT number (registered within clinicaltrials.gov). While the study objectives sound interesting, is important, and on target, a number of shortcomings were observed, in regards to abiding by the CONSORT guidelines for conducting and reporting results of high-quality randomized controlled trials (RCTs). Although statistical methods employed looked more or less adequate, some concerns remain:

1. Methods: Methods reporting require an orderly manner following CONSORT guidelines, without repeating information, such as Trial Design, Participant Eligibility criteria and settings, Interventions, Outcomes, sample size/power considerations, Interim analysis and stopping rules. Randomization (details on random number generation, allocation concealment, implementation), and Blinding considerations should be mentioned explicitly. The authors are advised to create separate subsections for each of the possible topics (whichever necessary), and that way produce a very clear writeup. I see the Authors already made a sincere attempt; however, they are advised to write it carefully, following nice examples in the manuscript below:

https://www.sciencedirect.com/science/article/pii/S0889540619300010

Specific comments below:

(a) For instance, the randomization and allocation concealment should be made very clear (they are NOT the same thing); the trial staff recruiting patients should NOT have the randomization list. Randomization should be prepared by the trial statistician, and he/she would not participate in the recruiting.

(b) Sample size/power: Power analysis is provided, but to dot clearly mention the "name of the statistical test" used. On what basis was the 2:2:1 allocation decision taken?

(c) Statistical Analysis:

(c1) Linear mixed models (with random intercepts and time) were used to model the PTSD severity; however, these models work well under Gaussian assumptions of the random terms (effects & errors). It is not clear if those assumptions were assessed, post model fitting. If those fail, alternative methods need to be employed.

(c2) Did the authors meant "Negative Binomial" regression, when they refereed to "Poisson Model with Overdispersion" for the DD & HDD?

2. Results:

The authors should check that any statement of significance should be followed by a p-value in the entire Results section. Otherwise, it looks OK.

3. Conclusions and Discussion:

Writeup should reflect that study findings from this RCT are based only on this specific population, and future studies (on other populations) are warranted to justify the hypothesized group differences.

Reviewer #2: This is an interesting results from a randomized trial that evaluated two active intervention compared to assessment only on PTSD symptoms, heavy drinking and drinking days in population diagnosed with PTSD and AUD. The individuals in the assessment only were then re-randomized to receive either CPT or relapse prevention and assessed for 12 months. This is a well-designed study with lots of advantages including the assessment of active treatment vs none then the comparison of two different active treatment. The hypothesis that one treatment for one disorder will be as good for the other disorder is novel. The assessment length for 12 months is another advantage. The main disadvantage that the author noted clearly in the manuscript is the sample size that did not reach the power size calculation. However, the manuscript is written with the confidence that any of these two treatment interventions is good for the other one. For example in the conclusion, and similar throughout the study, it says "Results indicate that treatments targeting one or the other aspect of the PTSD/AUD comorbidity have salutary

effects on both types of outcomes.". I suggest to change this to probable or possible because you cannot confirm there are equal if not reached the target sample size.

i would also like to see other information that is not presented in the study and if it was imbalanced , it might affect the results:

what was the percentage of people using antidepressnts in the three groups?

what is the percentage of people using anticraving for the three groups?

Was fidelity checked for the two active treatments? how?

What is the age at onset for PTSD and/or AUD in the three groups? If the trauma/PTSD happened after AUD (AUD-facilitated trauma) it might affect the individual perspective on what treatment is needed which might affect treatment retention. The opposite might be true.

In addition:

I am not clear what is the timeline for "post-treatment" " if both CPT and RP is 12 sessions then it should be three months. If so, what is the difference between post-treatment and 3-months timeline.

For example, line 197: how is it 20 weeks if it is 12 sessions?

Also:

line 376: " RP addressed PTSD symptoms that triggered craving or drinking," . How do you explain this?

The limitation talks a bit about it "the generic behavioral skills clients

446 gain through both CPT and RP (e.g., communication and assertiveness, distress tolerance,

447 reaching out for healthy social support)" but more theories can be introduced such as treating one disorder is automatically helping the other disorder. CPT could be just o PTSD which decrease the need for drinking. RP is more for drinking which alleviate the symptoms of PTSD. Thus, I would suggest to consider measuring craving as well in the future.

Finally, the fact that only about 50% each group completed treatment could be very likely related to the difficulty of recruitment which means that a lot of individuals with PTSD-AUD are not ready for trauma-focused therapy. This means that RP could actually have advantage for some individual to start with rather than to be sent to any available treatment. A lot of individuals might not tolerate a referral to CPT.

Reviewer #3: Pg 5, Ln 72: change “more acceptable patients” to “more acceptable to patients”

Pg 6, Ln 82: change “hypothesizes” to “hypotheses”

Exclusion critera: What is "unstable psychiatric mediation"? Just no dose changes?

Were the 3 clinicians practicing at both the university and VA sites?

Curious how other substance use disorders affected outcomes? Was this looked at as a covariate? If not, why?

Can you comment on the additional rapport-building time in AO and much improved adherence?

Can you comment more on sig improvement in PTSD and HDD in AO?

I'm very confused about no AEs being reported given that this was asked about at each visit. If participants were asked about AEs I find it difficult to believe that none of them ever reported having a cold or a headache, etc. Can you explain more how you ended up with no AEs reported?

Lastly, I'm curious, based on your charts it looks like RP reduces PTSD as much as CPT but reduces drinking more than CPT. Is there a basis for recommending RP over CPT for PTSD-AUD?

Thank you for putting land acknowledgment!

6. PLOS authors have the option to publish the peer review history of their article (what does this mean?). If published, this will include your full peer review and any attached files.

Reviewer #1: No

Reviewer #2: **Yes: **A. N. Hassan

Reviewer #3: No

---

## [Author Response · Author response to Decision Letter 0]

27 Jun 2022

Responses to Reviewers

Journal Requirements:

Response: We have corrected the formatting so that it now complies with PLOS ONE requirements.

"Dr. Kaysen is a co-author on a book on Cognitive Processing Therapy published by Elsevier for which she receives royalties. In addition she has conducted clinical workshops on Cognitive Processing Therapy for which she has received speakers fees, which could constitute a conflict of interest. Dr. Resick is a co-author on the Cognitive Processing Therapy treatment manual for which she receives royalties and she conducts clinical workshops on Cognitive Processing Therapy for which she receives speakers’ fees, which could constitute a conflict of interest. The other co-authors have no conflicts of interest to declare pertinent to this submission."

Response: Our updated competing interests are included in the revised cover letter. Additionally, this does not alter our adherence to PLOS ONE policies on sharing data and materials.

Response: We have uploaded the required files to the National Addiction and HIV Archive Program (NAHDAP; https://www.icpsr.umich.edu/web/pages/NAHDAP/index.html) and the Inter-university Consortium for Political and Social Research (ICPSR). At the present time an ICPSR data curator is reviewing the data and documentation in advance of approving the data collection for distribution on the ICPSR website and archiving the data for long-term preservation. Once the materials are vetted and the archiving is complete, we will be furnished with a doi number through which external users can access our data. We will forward that number to PLOS ONE once we receive it. 

Response: Our ethics statement only appears in the Methods section.

Response: We have added several citations to support revisions to the text and have double-checked all of the references for accuracy. To our knowledge, we are not using any references that have been retracted.

Additional Editor Comments:

This is an interesting manuscript. Comments from the reviewers should be addressed

Response: Thank you for the positive feedback. We have endeavored to address all of the reviewers’ comments below. The original reviewer comments are included followed by our responses with text insertions from the manuscript italicized.

Reviewers' comments:

Reviewer's Responses to Questions

Comments to the Author

1. Is the manuscript technically sound, and do the data support the conclusions?

Reviewer #1: Yes

Reviewer #2: Partly

Reviewer #3: Yes

2. Has the statistical analysis been performed appropriately and rigorously? 

Reviewer #1: Yes

Reviewer #2: Yes

Reviewer #3: Yes

3. Have the authors made all data underlying the findings in their manuscript fully available?

Reviewer #1: No

Reviewer #2: Yes

Reviewer #3: Yes

4. Is the manuscript presented in an intelligible fashion and written in standard English?

Reviewer #1: Yes

Reviewer #2: Yes

Reviewer #3: Yes

5. Review Comments to the Author

Reviewer #1: The objective of this RCT is to assess and compare the CPT treatment, an AUD treatment, and an assessment-only (AO) treatment, via a longitudinal study design. The study was approved by the respective IRB/Ethics Committee, and carries a valid NCT number (registered within clinicaltrials.gov). While the study objectives sound interesting, is important, and on target, a number of shortcomings were observed, in regards to abiding by the CONSORT guidelines for conducting and reporting results of high-quality randomized controlled trials (RCTs). Although statistical methods employed looked more or less adequate, some concerns remain:

1. Methods: Methods reporting require an orderly manner following CONSORT guidelines, without repeating information, such as Trial Design, Participant Eligibility criteria and settings, Interventions, Outcomes, sample size/power considerations, Interim analysis and stopping rules. Randomization (details on random number generation, allocation concealment, implementation), and Blinding considerations should be mentioned explicitly. The authors are advised to create separate subsections for each of the possible topics (whichever necessary), and that way produce a very clear writeup. I see the Authors already made a sincere attempt; however, they are advised to write it carefully, following nice examples in the manuscript below:

https://www.sciencedirect.com/science/article/pii/S0889540619300010

Response: Thank you for this guidance and for providing the helpful example. We have completely revamped the presentation of the Methods section starting on page 6 and it now conforms to the CONSORT guidelines, paying particular attention to delineating randomization from allocation.

Specific comments below:

(a) For instance, the randomization and allocation concealment should be made very clear (they are NOT the same thing); the trial staff recruiting patients should NOT have the randomization list. Randomization should be prepared by the trial statistician, and he/she would not participate in the recruiting.

Response: Thank you for pointing these issues out; we have now clarified that a priori stratified randomization tables were created by lead author Simpson who was not involved with participant recruitment or allocation. However, we now include in the limitations section that the Project Coordinator used the randomization tables to assign participants and was directly involved with participant recruitment and screening, which could have influenced allocation (pg. 31, lines 542-543).

(b) Sample size/power: Power analysis is provided, but to dot clearly mention the "name of the statistical test" used. On what basis was the 2:2:1 allocation decision taken?

Response: We have elaborated on the power calculations in the manuscript as follows (pg. 12, starting line 219): 

Considerable research shows that CPT and RP are effective treatments for their prescribed problems (i.e., PTSD for CPT and HDD for RP) and hence the study was powered to detect effects of each intervention on the non-prescribed targets (i.e., HDD for CPT and PTSD for RP). A priori power analyses were conducted via simulation using mixed effects generalized linear models229 with different outcome distributions (i.e., Normal vs. Poisson). Datasets were generated in which fixed effects for intercept and slope were based on preliminary studies and random-effects were generated based on random draws from a multivariate normal distribution (also specified based on prior research with PTSD and drinking outcomes). The simulation-based estimate of power was provided by the percentage of datasets in which the t statistic for the effect of intervention condition on change across time indicated significance at a critical value of p<.05. These procedures indicated a sample of 235 (CPT/RP each 95; AO 45) was needed to have .8 power to detect moderate (d = .4) effect sizes on the non-target outcomes for each active condition relative to the AO condition (i.e., drinking/CPT; PTSD/RP).

We have added the following text to explain our rationale for the 2:2:1 allocation scheme (pg. 13, starting line 242):

Because our primary interest was in the comparisons between participants assigned to CPT and to RP and to minimize the number of people whose active treatment receipt would be delayed, we opted to assign fewer participants to the AO condition.

(c) Statistical Analysis:

(c1) Linear mixed models (with random intercepts and time) were used to model the PTSD severity; however, these models work well under Gaussian assumptions of the random terms (effects & errors). It is not clear if those assumptions were assessed, post model fitting. If those fail, alternative methods need to be employed.

Response: With regard to the distribution of the measure of PTSD severity, the linear distributional model was chosen because at each time point absolute values for skewness were below .4 and absolute values for kurtosis were below .8. The limited deviation from a normal distribution in this study sample is not surprising given that all participants were required to meet DSM-5 diagnostic criteria for PTSD. With regard to distributions of random effects, we examine random effects graphically and calculated skewness and kurtosis of random effects for all models. In all cases, skewness was below an absolute value of .6. Kurtosis was also low (<1) for almost all random effects, with the exception of the random effects for post-tx time in models predicting PTSD symptom severity and HDD. Kurtosis for these random effects were 1.9 and 2.7, respectively, which are still below recommended cut-offs for violations of the normality assumption. The relatively high kurtosis values reflect the small amount of between-person variance in change across the post-tx time points. Although removing these random effects would have resulted in models that fit adequately and were more parsimonious (and yielded the same results with respect to fixed effects of treatment condition), we retained these random effects to be consistent with the study design and our research questions. We have added a sentence to the Methods indicating how we checked the normality assumption (pg. 16, starting line 293).

(c2) Did the authors meant "Negative Binomial" regression, when they refereed to "Poisson Model with Overdispersion" for the DD & HDD?

Response: Using the specifications available in HLM software for modeling count data in multilevel mixed models that contain random slopes in addition to random intercepts, we used Poisson models that include an over-dispersion term that makes the variance proportional to an observation’s expected value and accounts for the fact that variance of these outcomes exceeded their means. This approach also allows for estimation of random slopes as well as random intercepts. Multilevel negative binomial models, when implemented in most available software, only allow for random intercepts. The over-dispersed Poisson model is similar to a negative binomial model, which also includes an extra term and where the variance in the outcome is a quadratic function of the mean. This issue is described the bottom of page 7 and top of page 8 in the article by Atkins et al. (2013), and the formula for the variance term in the over-dispersed Poisson model is given in the HLM manual (Raudenbush et al., 2011). 

We made the following edits to the Methods to make the specification of these models clearer (pg. 15, starting line 283):

Over-dispersed Poisson models were used for DD and HDD (discrete non-negative integers showing positive skew; i.e., count outcomes). The over-dispersed Poisson model includes an extra term to account for the variance of outcomes exceeding the means (Atkins et al., 2013; Raudenbush et al., 2013), which was the case for the DD and HDD outcomes.

Atkins, D. C., Baldwin, S. A., Zheng, C., Gallop, R. J., & Neighbors, C. (2013). A tutorial on count regression and zero-altered count models for longitudinal substance use data. Psychology of Addictive Behaviors, 27(1), 166.

Raudenbush, S. W., Bryk, A.S., Cheong, Y.F., Congdon, R.T., & du Toit, M. HLM 7: Hierarchical linear and nonlinear modeling. Scientific Software International; 2011.

2. Results:

The authors should check that any statement of significance should be followed by a p-value in the entire Results section. Otherwise, it looks OK.

Response: p-values have been added throughout the Results section.

3. Conclusions and Discussion:

Write-up should reflect that study findings from this RCT are based only on this specific population, and future studies (on other populations) are warranted to justify the hypothesized group differences.

Response: Thank you for this suggestion; it is in line with comments from Reviewer 2. In response we have tempered our conclusions and altered the following sentence in the Conclusion section (here, italics denote wording changes) (pg. 32, starting line 565):

Results from this trial suggest that treatments targeting one or the other aspect of the PTSD/AUD comorbidity may have salutary effects on both types of outcomes.

Reviewer #2: This is an interesting results from a randomized trial that evaluated two active intervention compared to assessment only on PTSD symptoms, heavy drinking and drinking days in population diagnosed with PTSD and AUD. The individuals in the assessment only were then re-randomized to receive either CPT or relapse prevention and assessed for 12 months. This is a well-designed study with lots of advantages including the assessment of active treatment vs none then the comparison of two different active treatment. The hypothesis that one treatment for one disorder will be as good for the other disorder is novel. The assessment length for 12 months is another advantage. The main disadvantage that the author noted clearly in the manuscript is the sample size that did not reach the power size calculation. However, the manuscript is written with the confidence that any of these two treatment interventions is good for the other one. For example in the conclusion, and similar throughout the study, it says "Results indicate that treatments targeting one or the other aspect of the PTSD/AUD comorbidity have salutary effects on both types of outcomes." I suggest to change this to probable or possible because you cannot confirm there are equal if not reached the target sample size.

Response: As noted above in response to similar comments from Reviewer 1, we edited the noted sentence to be more circumspect and tentative given the small sample size (pg. 32, starting line 565).

I would also like to see other information that is not presented in the study and if it was imbalanced, it might affect the results:

what was the percentage of people using antidepressants in the three groups?

what is the percentage of people using anticraving for the three groups?

Response: Unfortunately while we recorded the details of study participants’ medication regimens in their baseline case report forms, we did not enter those details into our database so we cannot speak to the specifics of receipt of either antidepressants or anti-craving medications. The hard copies of the forms are now archived off-site and we do not have a timely way to retrieve them nor the staff to assist with data entry. Fortunately, however, we do not see evidence of a condition imbalance with regard to psychiatric medications generally, as reflected in online Table S3.

Was fidelity checked for the two active treatments? how?

Response: Yes, we checked for treatment fidelity by evaluating session audiotapes by a trained rater (please see the Intervention sub-section in the Methods section; pg. 10, starting line 168).

What is the age at onset for PTSD and/or AUD in the three groups? If the trauma/PTSD happened after AUD (AUD-facilitated trauma) it might affect the individual perspective on what treatment is needed which might affect treatment retention. The opposite might be true.

Response: We now include the age of onset of participants’ index traumas and problem(s) with alcohol in online Table S3 by assigned condition as well as the proportions in each group that experienced the index trauma first, alcohol problem(s) first, or at the same age. 

Although we found that those assigned to CPT experienced their index traumas at older ages than those assigned to RP and were more likely to report onset of alcohol problems prior to their index traumas, we have opted to simply share these observations in the participant characteristics section of the results rather than delve into how these onset patterns may or may not be associated with treatment retention or outcomes as this deserves more space and attention than can be afforded in the primary outcome paper.

In addition:

I am not clear what is the timeline for "post-treatment" " if both CPT and RP is 12 sessions then it should be three months. If so, what is the difference between post-treatment and 3-months timeline.

Response: We appreciate that the assessment timing is somewhat confusing and have attempted to clarify it with the following new text (pg. 12, starting line 204):

Because people were allowed 20 weeks in which to complete the 12-session interventions baseline to the first post-treatment assessment was somewhat variable, but the 3- and 12-month assessments took place 3 and 12 months after treatment ended.

For example, line 197: how is it 20 weeks if it is 12 sessions?

Response: Many of our participants struggled to attend treatment sessions consistently due to various life challenges (e.g., over-sleeping, getting a job for the day, having childcare fall through, etc.). Thus we extended the treatment window to 20 weeks so as to afford them a reasonably sufficient time in which to obtain as much treatment as they could without having their active intervention engagement drag on indefinitely. We have clarified this in the methods section as follows (italics denote new text) (pg. 9, starting line 153):

“Participants attended individual therapy sessions once or twice per week for a maximum of 20 weeks. The study was originally designed such that participants initially assigned to an active treatment would attend twice weekly sessions over 6-weeks. However, this treatment schedule proved too challenging for most participants and the treatment window was adjusted to allow up to 20 weeks for treatment completion. Thus, the treatment duration during the active treatment phase was 0 to 20 weeks.”

The limitation talks a bit about it "the generic behavioral skills clients gain through both CPT and RP (e.g., communication and assertiveness, distress tolerance, reaching out for healthy social support)" but more theories can be introduced such as treating one disorder is automatically helping the other disorder. CPT could be just o PTSD which decrease the need for drinking. RP is more for drinking which alleviate the symptoms of PTSD. Thus, I would suggest to consider measuring craving as well in the future.

Response: Yes, we absolutely agree that the interventions could be having positive effects on their target outcomes (CPT on PTSD and RP on drinking), which in turn led to positive effects on the non-target outcomes. We have included new language describing this possibility in the discussion (pg. 30, starting line 521). 

It is possible that general behavioral skills clients gain through both CPT and RP (e.g., improved communication and assertiveness, distress tolerance, reaching out for healthy social support, coping self-efficacy) are helpful in addressing underlying dimensions of psychopathology common to both SUD and PTSD. Gaining such skills may help clients mitigate current stress, better tolerate negative emotions, and temper stress reactivity [69–72], including those associated with cravings in response to substance or trauma-related cues. It is also possible that improvement in one or the other set of signs and symptoms (i.e., PTSD or AUD) leads to salutary effects in the other. Indeed, support for the idea that reductions in PTSD are associated with later reductions in substance use was found [73] in a large behavioral treatment trial involving women with co-occurring PTSD and SUD. This secondary analysis did not, however, find that reductions in substance use were associated with later improvements in PTSD. Future research on both the sequence of symptom change and mechanisms of change associated with these two interventions will be important for unraveling the finding that they are each associated not only with positive changes in their target outcomes, but also in their non-target outcomes. 

With regard to the suggestion about assessing cravings, we did include a measure of cravings and found the same pattern for craving as the drinking outcomes; no between group differences for the CPT and RP comparisons. Given that the paper is already long and complex with multiple primary outcomes included, we prefer not to add craving but appreciate the reviewer’s suggestion to consider this indicator of alcohol involvement.

Finally, the fact that only about 50% of each group completed treatment could be very likely related to the difficulty of recruitment which means that a lot of individuals with PTSD-AUD are not ready for trauma-focused therapy. This means that RP could actually have advantage for some individual to start with rather than to be sent to any available treatment. A lot of individuals might not tolerate a referral to CPT.

Response: Although it is true that treatment completion rates were low in the present study, these same low rates were seen nearly across the board in a recently published meta-analysis conducted by members of our team (Simpson et al., 2021). It may be that people with PTSD/AUD are reluctant to engage with treatment in the first place or may have difficulty staying in treatment because they are concerned that trauma-focused treatment will be too challenging, but in our study and in the larger literature, the treatment completion rates are also quite low for non-trauma focused interventions, including manualized SUD care like RP, or treatments like Seeking Safety. These overall findings suggest that people with PTSD/AUD (or PTSD/SUD more generally) have an especially hard time engaging in and sticking with treatment of any sort and future research is needed to ascertain why this is and whether it has to do with the treatments we currently have available for them or the life circumstances they are contending with (or both).

Reviewer #3: 

Pg 5, Ln 72: change “more acceptable patients” to “more acceptable to patients”

 Response: Thank you for pointing out this error; it has been corrected.

Pg 6, Ln 82: change “hypothesizes” to “hypotheses”

Response: Thank you for pointing out this error as well; it has been corrected.

Exclusion criteria: What is "unstable psychiatric mediation"? Just no dose changes?

Response: We now note in the Methods section that “unstable psychiatric medication” refers both to dose changes and to the addition/subtraction of medications (see pg. 8, starting line 119).

Were the 3 clinicians practicing at both the university and VA sites?

Response: We now note that two of the three clinicians provided care at the university clinic site and that the third provided care at the VA clinic site (see pg. 10, starting line 165).

Curious how other substance use disorders affected outcomes? Was this looked at as a covariate? If not, why?

Response: Because there were no differences between randomized study conditions with regard to additional drug use disorders or use of commonly used drugs, we did not control for this in the models for parsimony.

Can you comment on the additional rapport-building time in AO and much improved adherence?

Response: Although those initially assigned to AO were more likely to participate in the first post-baseline assessment than those randomized to receive either CPT or RP right away, the AO group did not have better retention or adherence when it came to treatment completion. As may be seen in the table below, if anything, those who were initially assigned to the AO condition had slightly worse treatment retention than the other two groups. Completion of 9 or more sessions did not differ between those who were and were not initially assigned to AO, χ2(1) = 0.34, p = .561. Because there does not appear to be an effect of the pre-treatment support calls on later treatment retention, we have not included this information in the revised manuscript although it generally does not support the idea that additional rapport building improved adherence.

 Treatment Condition

 AO then CPT AO then RP CPT RP

Completed 9 sessions? no Count 6 5 19 15

 % within treatment condition 50.0% 50.0% 46.3% 39.5%

 yes Count 6 5 22 23

 % within treatment condition 50.0% 50.0% 53.7% 60.5%

Total Count 12 10 41 38

 % within treatment condition 100.0% 100.0% 100.0% 100.0%

Can you comment more on sig improvement in PTSD and HDD in AO?

Response: Thank you for the opportunity to speak to this particular finding. We have added the following to the Discussion section to address it (pg. 26, starting line 439):

Although receipt of an active intervention was more beneficial than assignment to AO, it is noteworthy that those assigned to the AO condition made significant within-subject improvements in both their PTSD symptoms and their HDD. These findings are consistent with other research showing that many people reduce their drinking after making the decision to seek care but prior to actual receipt of care (Stasiewicsz et al. 2019), and that many people with PTSD benefit from non-directive phone support (Lehavot et al., 2021).

I'm very confused about no AEs being reported given that this was asked about at each visit. If participants were asked about AEs I find it difficult to believe that none of them ever reported having a cold or a headache, etc. Can you explain more how you ended up with no AEs reported?

Response: Thank you for flagging this important issue. As you point out, participants certainly reported distress over the course of the study. Many of our study participants experienced PTSD exacerbations, reported suicidal ideation, and at times consumed large quantities of alcohol; all of these challenges were anticipated (i.e., they are common regardless of treatment and were expected) for this group of patients as determined by both IRBs and NIAAA, the study sponsor. Adverse events are typically defined as unexpected medical or psychiatric problems during treatment. Thus, our original statement that there were no AE's stands. We have, however, noted the challenges that our study participants most frequently experienced and clarified that they were anticipated. (see pg. 25, starting line 409). 

Lastly, I'm curious, based on your charts it looks like RP reduces PTSD as much as CPT but reduces drinking more than CPT. Is there a basis for recommending RP over CPT for PTSD-AUD?

Response: Although we appreciate the Reviewer’s observations regarding the comparable PTSD outcomes for those assigned to CPT and to RP and the better heavy drinking outcomes for the latter group, because we saw evidence that CPT was more helpful for PTSD relative to the AO condition early on and RP was not, we are reluctant to recommend RP over CPT at this point.

Thank you for putting land acknowledgment!

 Response: You are most welcome and thank you for the reinforcement!

6. PLOS authors have the option to publish the peer review history of their article (what does this mean?). If published, this will include your full peer review and any attached files.

Do you want your identity to be public for this peer review? For information about this choice, including consent withdrawal, please see our Privacy Policy.

Reviewer #1: No

Reviewer #2: Yes: A. N. Hassan

Reviewer #3: No

---

## [Decision Letter · Decision Letter 1]

29 Sep 2022

Cognitive Processing Therapy or Relapse Prevention for comorbid Posttraumatic Stress Disorder and Alcohol Use Disorder: A randomized clinical trial

PONE-D-22-03482R1

Dear Dr. Simpson,

We’re pleased to inform you that your manuscript has been judged scientifically suitable for publication and will be formally accepted for publication once it meets all outstanding technical requirements.

Kind regards,

Dario Ummarino, PhD

Senior Editor

PLOS ONE

Reviewers' comments:

Reviewer's Responses to Questions

**Comments to the Author**

1. If the authors have adequately addressed your comments raised in a previous round of review and you feel that this manuscript is now acceptable for publication, you may indicate that here to bypass the “Comments to the Author” section, enter your conflict of interest statement in the “Confidential to Editor” section, and submit your "Accept" recommendation.

Reviewer #1: All comments have been addressed

Reviewer #3: All comments have been addressed

2. Is the manuscript technically sound, and do the data support the conclusions?

Reviewer #1: (No Response)

Reviewer #3: Yes

3. Has the statistical analysis been performed appropriately and rigorously? 

Reviewer #1: (No Response)

Reviewer #3: I Don't Know

4. Have the authors made all data underlying the findings in their manuscript fully available?

Reviewer #1: (No Response)

Reviewer #3: Yes

5. Is the manuscript presented in an intelligible fashion and written in standard English?

Reviewer #1: (No Response)

Reviewer #3: Yes

6. Review Comments to the Author

Reviewer #1: (No Response)

Reviewer #3: (No Response)

7. PLOS authors have the option to publish the peer review history of their article (what does this mean?). If published, this will include your full peer review and any attached files.

Reviewer #1: No

Reviewer #3: No

---

## [Editor Report · Acceptance letter]

17 Nov 2022

PONE-D-22-03482R1 

Cognitive Processing Therapy or Relapse Prevention for comorbid Posttraumatic Stress Disorder and Alcohol Use Disorder: A randomized clinical trial 

Dear Dr. Simpson:

I'm pleased to inform you that your manuscript has been deemed suitable for publication in PLOS ONE. Congratulations! Your manuscript is now with our production department. 

Kind regards, 

on behalf of

Dr Dario Ummarino, PhD 

Staff Editor

PLOS ONE